# HEAT UP THE SENTIMENT LEARNING WITH ICE

**Yao Yao[1,2], Zuchao Li[3,*] and Hai Zhao[1,2,*]**
[1]Department of Computer Science and Engineering, Shanghai Jiao Tong University
[2]MoE Key Lab of Artificial Intelligence, AI Institute, Shanghai Jiao Tong University
[3]National Engineering Research Center for Multimedia Software,
School of Computer Science, Wuhan University, Wuhan, 430072, P. R. China
`yaoyao27@sjtu.edu.cn`, `zcli-charlie@whu.edu.cn`,
`zhaohai@cs.sjtu.edu.cn`

## ABSTRACT

Recently, dramatic gains have been made on the task of aspect sentiment triplet extraction (ASTE). In this paper, we introduce a straightforward pipeline model to perform two-stage sequence labeling, including aspect and opinion terms identification and aspect-opinion pair classification. To exploit the cross-sentence context information to the maximum extent possible, we propose the instance cooperative enhancement (ICE) by introducing unsupervised clustering methods. Through experimenting with various clustering methods, we found that GSDMM unleashes the potential of cross-sentence information to the most degree. Compared to current state-of-the-art models, the results show the effectiveness of our proposed framework on ASTE-Data-V2.

## 1 INTRODUCTION

ASTE aims at extracting aspect sentiment triplets in a sentence and each triplet is composed of an aspect term, an opinion term, and their sentiment dependency. An example of ASTE can be found in Appendix A. However, the recent work Zhong & Chen (2021) discovered that the contextual representations for the entity and relation models essentially capture distinct information. Inspired by them, we restore ASTE from joint model Xu et al. (2020) to pipeline model Li et al. (2019); Wang et al. (2017) and return to the original nature of ASTE by introducing a straightforward pipeline model for ASTE.

We also innovatively introduce instance cooperative enhancement (ICE) to help our model infiltrate the input with cross-sentence context. ICE allows instances to cooperate with each other and hence help our pre-trained encoder generate more specific representations. To be more specific, we first apply a collapsed Gibbs Sampling algorithm for the Dirichlet Multinomial Mixture model (GSDMM) Yin & Wang (2014) to cluster similar sentences into groups. Then we concatenate the input sentence with cross-sentence context information within the same group. Many previous works proved that incorporating cross-sentence information can help model generate more accurate representations Wadden et al. (2019); Zhong & Chen (2021); Luan et al. (2019). To unleash the utmost potential of utilizing cross-sentence information, we conduct our experiments on various clustering methods, including simple clustering, k-means and GSDMM. We found that GSDMM shows great advantage in instance cooperative enhancement and effectively cope with the long sentence dependency problem.

## 2 METHOD

Inspired by Zhong & Chen (2021), Our proposed framework for ASTE consists of the following steps: 1) Instance cooperative enhancement: enhance the input sentence with cross-sentence context information. 2) Aspect and opinion identification: predict the aspect terms and opinion terms within the input sentence. 3) Aspect-opinion pair classification: predict the relation type between given aspect-opinion pair which is highlighted by special tokens `<S:label>`, `</S:label>`, `<O:label>` and `</O:label>`. The overview of our framework can be seen in Figure 1. The

---

* Corresponding author.

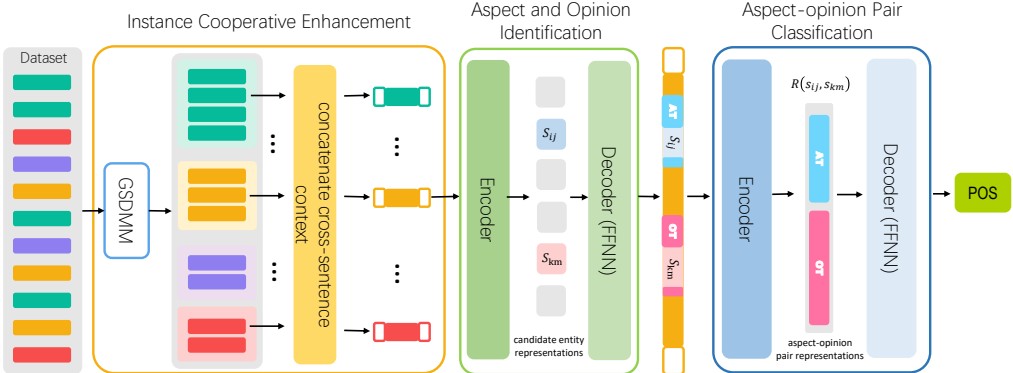

Figure 1: Overview of ICE pipeline model for aspect sentiment term extraction

model at each stage is trained independently, and each model only relies on the output features provided by the previous stage. The detailed formulation of our Two Stage ASTE model can be found in Appendix C and experimental settings can be found in Appendix D

## 2.1 Instance Cooperative Enhancement

We introduce instance cooperative enhancement (ICE) to incorporate global context for the input sentence. Specifically, our model fixes the length of the input sentence to $L$. When given a sentence with a length of $l$, we concatenate the sentence with $\frac{L-l}{2}$ left and right context within the same group respectively. Whereas, ASTE dataset is created on sentence-level instead of document-level which makes it difficult for us to utilize cross-sentence context. Therefore, we employ different text clustering methods to cluster similar sentences into groups. We experiment the following clustering methods respectively:

• **Simple Cluster** Since the ASTE dataset is originally extracted from restaurant and laptop reviews and ordered by these two domains. We employed a simple cluster method which clusters five adjacent sentences into one group.

• **K-means** We also employ K-means MacQueen (1967) using cosine similarity for text clustering.

• **GSDMM** The collapsed Gibbs Sampling algorithm for the Dirichlet Multinomial Mixture model (GSDMM) is a short text clustering method proposed by Yin & Wang (2014). A simple analogy model and algorithm to explain the GSDMM can be found in Appendix B

## 3 Results and Analysis

For ASTE task, our model achieves a new state-of-the-art in all four datasets of ASTE-Data-V2. By conducting experiments on all three clustering methods, we can see from Table 1 that in most of cases, GSDMM greatly outperforms simple clustering and K-means. Comparing to K-means, GSDMM provides a more stable improvement on all four datasets. According to the algorithm , we believe the main reason is that instead of previously defining the number of clusters as K-means did, GSDMM Yin & Wang (2014) can automatically calculate the number of clustered documents with a better balance between the completeness and homogeneity of the clustering results and hence wield a more stable and significant improvement. Further explorations on ICE and performance on different sentence lengths can be found in Appendix E

| Model | 14lap | | | | 14res | | | | 15res | | | | 16res | | | |
|---|---|---|---|---|---|---|---|---|---|---|---|---|---|---|---|---|
| | P | R | F1 | Δ | P | R | F1 | Δ | P | R | F1 | Δ | P | R | F1 | Δ |
| Xu et al. (2020)+BERT | 55.39 | 47.33 | 51.04 | | 70.56 | 55.94 | 62.40 | | 64.45 | 51.96 | 57.53 | | 70.42 | 58.37 | 63.83 | |
| Yan et al. (2021)+BART | 58.69 | 56.19 | 58.69 | | 65.52 | 64.99 | 65.25 | | 59.14 | 59.38 | 59.26 | | 66.60 | 68.68 | 67.62 | |
| Xu et al. (2021)+BERT | 63.44 | 55.84 | 59.38 | | 72.89 | 70.89 | 71.85 | | 62.18 | 64.45 | 63.27 | | 69.45 | 71.17 | 70.26 | |
| Zhang et al. (2021)+BERT | - | - | 60.78 | | - | - | 72.16 | | - | - | 62.10 | | - | - | 70.10 | |
| Ours(Simple)+BERT | 65.58 | 56.01 | 60.42 | - | 72.15 | 68.81 | 70.44 | - | 67.59 | 60.21 | 63.69 | - | 71.51 | 69.84 | 70.67 | - |
| Ours(Kmeans)+BERT | 67.41 | 55.82 | **61.07** | +0.65 | 72.60 | 69.32 | 70.92 | +0.48 | 64.68 | 62.68 | 63.66 | -0.03 | 73.94 | 67.90 | 70.79 | +0.12 |
| Ours(GSDMM)+BERT | 63.49 | 57.86 | 60.54 | +0.12 | 72.23 | 72.23 | **72.23** | +1.79 | 66.45 | 62.47 | **64.40** | +0.71 | 75.91 | 69.26 | **72.43** | +1.76 |

Table 1: The overall results for ASTE task.

## 4  CONCLUSION

In this work, we propose a straightforward pipeline model with instance cooperative enhancement for aspect sentiment triplet extraction. To probe deep into the importance of cross-sentence context information, we experiment on various clustering methods and conclude that GSDMM can unleash the potential of context information to the most degree. We prove that ICE can also effectively solve the long-distance dependency problem. Without using any extrinsic information such as syntax, our model reaches competitive results on ASTE-Data-V2 dataset.

## URM STATEMENT

The authors acknowledge that at least one key author of this work meets the URM criteria of ICLR 2023 Tiny Papers Track.

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

## A ASTE EXAMPLE

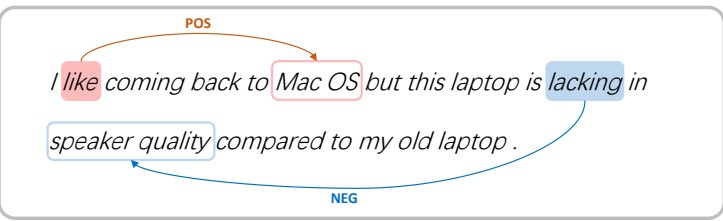

Figure 2: Examples for ASTE

## B GSDMM

The authors Yin & Wang (2014) introduce a simple analogy model to explain the GSDMM called the Movie Group Process. Imagine that a professor starts a movie discussion. Before the class begins, the students make lists of their favorite films and are first randomly assigned to K tables. The professor decides to cluster the students into several groups. Students in the same group will share similar interests and vice versa. The professor then asks each student to re-join a table and the student are expected to choose a new table according to the following two rules: (1) the new table should have more student; (2) The students of the new table should watch more similar movie with him. By following these steps iteratively, the students would eventually fall into stable clusters.

The detailed algorithm is shown in Algorithm 1 where the $p(T_s = t|\mathbf{T}_{\neg}s, s)$ was derived from the Dirichlet Multinomial Mixture (DMM) model Nigam et al. (2000) and can be represented as follows:

$$p(T_s = t|\mathbf{T}_{\neg}s, s) \propto$$

$$\frac{N_{d,\neg s} + \alpha}{S - 1 + K\alpha} \frac{\prod_{w \in s} \left( o_{d,\neg s}^w + \beta \right)}{\prod_{i=1}^{n_s} \left( n_{d,\neg s} + V\beta + i - 1 \right)} \tag{1}$$

where $S$ is the number of sentences in the dataset; $n_s$ is the number of words in sentence $s$; $V$ is the number of words in the vocabulary; $\alpha$ and $\beta$ are hyperparameters which respectively control the probability of a sentence being clustered to an empty group and the probability of the sentence choosing a group with more similar sentences. We can see that Equation 1 follows two rules of the Movie Group Process model. Specifically, the first part follows the first rule where a sentence would be assigned to a group with more sentences and the second part of the Equation 1 satisfied the second rule where sentence will be clustered to a group which share similar words with it.

---

**Algorithm 1** GSDMM

---

**Annotation:** $S$: sentences; $N_d$: number of sentences in group $d$; $n_s$: number of words in sentence $s$; $o_d^w$: number of occurrences of word $w$ in group $d$; $\mathbf{T}$: group labels for each sentences; $\mathbf{T}_{\neg}s$: $\mathbf{T}$ without sentence $s$ group label

**Input:** $\mathbf{S}$: input sentences

**Output:** $\mathbf{T} = \{T_1, T_2...T_n\}$; $T_s$ is the group label of sentences $s$

1: **for** group $d \in D$ **do**
2:      initialize $N_d \leftarrow 0, n_d \leftarrow 0, o_d^w \leftarrow 0$ ;
3: **end for**
4: **for** $s \in \mathbf{S}$ **do**
5:      sample sentence $s$ a group
6:      $T_s \leftarrow t \sim Multinomial(1/K)$;
7:      update group properties
8:      $N_d \leftarrow N_d + 1; n_d \leftarrow n_d + o_s$
9:      **for** every word $w \in s$ **do**
10:          $o_d^w \leftarrow o_d^w + N_s^w$
11:      **end for**
12: **end for**
13: **for** iteration $i \in [1, I]$ **do**
14:      **for** every sentence $s \in \mathbf{S}$ **do**
15:          record the current group of sentence $s$: $t \leftarrow t_s$
16:          remove $s$ from group $d$ and update the group properties:
17:          $N_d \leftarrow N_d - 1; n_d \leftarrow n_d - o_s$
18:          **for** every word $w \in s$ **do**
19:              $o_d^w \leftarrow o_d^w - N_s^w$
20:          **end for**
21:          sample a new group for $s$
22:          $T_s \leftarrow t \sim p(T_s = t | \mathbf{T}_{\neg}s, s)$
23:          $N_d \leftarrow N_d + 1; n_d \leftarrow n_d + o_s$
24:          **for** every word $w \in s$ **do**
25:              $o_d^w \leftarrow o_d^w + N_s^w$
26:          **end for**
27:      **end for**
28: **end for**

---

## C    Two Stage ASTE model

Following Zhong & Chen (2021), our model consists of two stage (1) Aspect and opinion identification; (2) Aspect-opinion pair classification.

### C.0.1   Aspect and Opinion Identification

In the first stage, our model aims at predicting the aspect terms and opinion terms of the input sentence $X = \{x_1, x_2, ..., x_n\}$. We first build contextualized representations $H = \{h_1, h_2, ..., h_n\}$ for the input sentence using pretrained language model. For every candidate span with length less than $L$, we construct its span representation:

$$s_{ij} = [h_i; h_j; \phi(i, j)] \qquad (2)$$

where $\phi(i, j)$ is the distance feature vector which is used to encode the span width. We then use a feedforward neural network as a decoder to classify each span. The probability distribution of span types is calculated by:

$$p(y|s_{ij}) = softmax(\mathbf{FFNN}(s_{ij})) \qquad (3)$$

where $y \in Y = \{AT, OT, \epsilon\}$

### C.0.2   Aspect-opinion Pair Classification

In the pair classification model, we first insert special tokens to highlight aspect and opinion terms predicted by the first stage. For instance, for spans $s_{ij}$ and $s_{km}$ with the labels of $y_i$ and $y_k \in Y$,

our highlighted input sentence $X$ can be denoted as:

$$X = \{x_1, ..., < s : y_i >, x_i, ..., x_j, < /s : y_i >, ...,$$
$$< o : y_k >, x_k, ..., x_m, < /s : y_k >, ..., x_n\} \quad (4)$$

We then use another encoder to build contextualized representations $H$ for highlighted input $X$.

$$H = \{..., h_i^s, h_i, ..., h_j, h_i^e, ..., h_k^s, h_k, ..., h_m, h_k^e, ...\} \quad (5)$$

The relation pair representation of span $s_{ij}$ and $s_{km}$ is calculated by :

$$R(s_{ij}, s_{km}) = [h_i^s, h_k^s] \quad (6)$$

Similar to stage 1, we also use a feedforward neural network to calculate the probability distribution of relation types.

$$p(y_{rel}|(s_{ij}, s_{km}) = softmax(\mathbf{FFNN}(R(s_{ij}, s_{km}))) \quad (7)$$

where $y_{rel} \in Y_{rel} = \{POS, NEG, NEU, \epsilon\}$

## D  EXPERIMENTS

We experiment our model on ASTE-Data-V2 dataset Xu et al. (2020). Our models are implemented based on HuggingFace's Transformers library Wolf et al. (2019). For all stages in our pipelined model, we use the pre-trained language model bert-based-uncased Devlin et al. (2018) as the base encoders, and all two models are trained for 100 epochs, and with a learning rate of 1e-5, a batch size of 16, and a dropout probability of 0.1 for the classifier. We fix the length of the input sentence to $L = 80$ using instance cooperative enhancement and for GSDMM, we set $K = 121, \alpha = 0.1, \beta = 0.1$ and train the model for 100 iterations. We evaluate our model with all three cluster methods.

## E  FURTHER EXPLORATION

### E.1  INSTANCE COOPERATIVE ENHANCEMENT OR NOT?

|  | 14lap | | | |
|---|---|---|---|---|
|  | P | R | F1 | Delta |
| Ours -wo ICE(L=0) | 58.02 | 57.49 | 57.75 | - |
| -w ICE(L=70) | 63.06 | 57.12 | 59.94 | +2.19 |
| -w ICE(L=80) | 65.58 | 56.01 | **60.42** | +2.67 |
| -w ICE(L=90) | 65.49 | 55.08 | 59.84 | +2.09 |
| -w ICE(L=100) | 66.59 | 53.79 | 59.51 | +1.76 |
| -w ICE(L=200) | 63.75 | 55.27 | 59.21 | +1.46 |

Table 2: Comparison between with (-w) or without (-wo) ICE

To prove that the instance cooperative enhancement (ICE) efficiently improves the overall performance, we use original sentences as input without concatenating the left and right context within the same document. To further probe into the influence ICE exerts on the results, we experiment the ICE with different input length $L$. The results are shown in Table 2. We can see that ICE is of great importance for improvement of the model performance. Models with ICE greatly outperforms those without ICE by a large margin of 2.67 when $L$ is set to 80. We find that the F1 score reaches the peak value when $L = 80$ and do not further increases when the input length grows. We believe that adding cross-sentence information on one hand can help our model capture deeper relations between entities, but on the other hand, too much cross-sentence information enhancement also introduces noises and hence hurts the overall performance. Therefore, we use $L = 80$ for our ASTE experiments.

### E.2  PERFORMANCE ON DIFFERENT SENTENCE LENGTHS

Since we employ ICE to incorporate richer context information, we speculate that the most performance improvement from our model may attribute to its solved long-distance dependency problem.

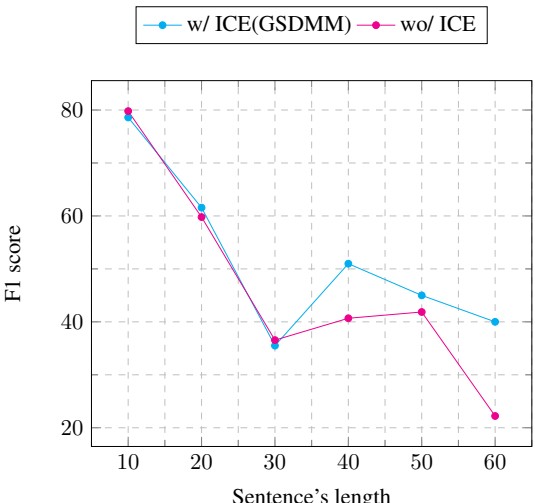

Figure 3: Performance of different length ranges in 14lap dataset.

To verify our hypothesis, we deprive our model of ICE and compare it with our full model. We perform their statistics on the 14lap dataset for different length ranges, and the results are shown in Figure 3.

According to the curves in the figure, it is obvious that the deprived model and the full model shares similar results in short sentences (< 30), while when not using ICE, the model suffers a significant decrease as the sentence length grows. In contrast, our model has the smallest decrease and obtains the best results, thus indicating that our method can effectively cope with the long sentence dependency problem.

