# OpenReview forum: "Heat Up The Sentiment Learning With ICE"
_ICLR.cc/2023/TinyPapers — Submitted to Tiny Papers @ ICLR 2023_

### Official Review · Reviewer_2vG3 · 2023-03-20

**Confidence:** 3

**Summary Of Contributions:**

An interesting paper for aspect sentiment triplet extraction

**Rating:**

Great Start (GS): a submission which meets some of the reviewing criteria but has room for improvement

**Strengths And Weaknesses:**

This paper proposes the Instance Cooperative Enhancement (ICE) model, which incorporates unsupervised clustering methods. The experiments with various clustering methods demonstrate that GSDMM maximizes the potential of cross-sentence information. Compared to current state-of-the-art models, the results demonstrate the effectiveness of the proposed framework on the ASTE-Data-V2 dataset.

Strengths:

1.The paper introduces a new model, the Instance Cooperative Enhancement (ICE), which utilizes unsupervised clustering methods.

2.The experimental results demonstrate the effectiveness of the proposed model.

Weaknesses:

1.The paper's novelty is somewhat limited, which is a concern.

2.The experimental improvement is relatively small.

**Suggested Changes:**

1.The paper's novelty is somewhat limited, which is a concern.

2.The experimental improvement is relatively small.

---

### Official Review · Reviewer_DfNP · 2023-03-24

**Confidence:** 4

**Summary Of Contributions:**

Author/s presents unsupervised methods for sentiment analysis by performing three important task on the dataset Instance Cooperative Enhancement, Aspect and Opinion Identification and lastly Aspect-Opinion Classification. The Author/s perform grouping of similar sentences from the dataset, then they identify aspects and opinion out of the grouped sentences where afterward they combine the aspect and opinion to perform overall sentiment of a sentence.

**Rating:**

Clear, Correct, and Reproducible (CCR): a submission which meets the reviewing criteria

**Strengths And Weaknesses:**

Strength:
* the paper provide solution to sentences where a sentence contains a mix of positive, negative where through the solution provided they suggest they can classify the sentence.\
Weakness:\
the following is rather the questions that might be answered such as
* is the study really unsupervised machine learning since there were a mention of some of the techniques performing better than others a better explanation of metrics used would shade more light.
* In appendices there is a mention of F1 score, I wonder about the dataset used and algorithm used where there were deep learning approaches such sequence to sequence models where I wonder which machine learning type used in the study?

**Suggested Changes:**

ICE initials in the paper thought it should be in capital where most the time they were instance cooperative
enhancement (ICE)
suggested changes:
Instance Cooperative Enhancement (ICE)

---

### Meta-Review · Area_Chair_T9tC · 2023-04-04

**Recommendation:** Invite to present
**Confidence:** 4

**Metareview:**

Good paper with all reviewers arguing for acceptance. Some changes are suggested.

**Summary:**

In this study, the Instance Cooperative Enhancement (ICE) model is proposed, integrating unsupervised clustering techniques. The authors are suggested to modify the manuscript according to the questions/concerns from the review.

**Reason For Not Giving A Higher Recommendation:**

Some clarifications are needed on the methodological and empirical settings.

**Reason For Not Giving A Lower Recommendation:**

N/A

---

### Decision · Program_Chairs · 2023-04-10

Invite to present